# A Proposal for the Application of Failure Assessment Diagrams to Subcritical Hydrogen Induced Cracking Propagation Processes

**Borja Arroyo Martínez ***, **José Alberto Álvarez Laso, Federico Gutiérrez-Solana,**
**Alberto Cayón Martínez, Yahoska Julieth Jirón Martínez and Ana Ruht Seco Aparicio**

LADICIM (Laboratory of Materials Science and Engineering), University of Cantabria, ETS. Ingenieros de Caminos, Av/Los Castros 44, Santander 39005, Spain; alvareja@unican.es (J.A.Á.L.); gsolana@unican.es (F.G.-S.); acayon@adif.es (A.C.M.); juliethjiron0492@gmail.com (Y.J.J.M.); anaruthseco@hotmail.com (A.R.S.A.)
* Correspondence: arroyob@unican.es; Tel.: +34-942-201837

**Abstract:** In this work, an optimization proposal for a model based on the definition of regions for crack propagation by means of the micromechanical comparison by SEM images and its application to failure assessment diagrams (FADs) is presented. It consists in three approaches. (1) The definition of the crack propagation initiation in the elastic-plastic range. (2) A slight modification of the zones in which the FAD is divided for hydrogen induced cracking (HIC) conditions. (3) The introduction of a simple correction for the definition of the $K_r$ coordinate of the FAD to take into account the fracture toughness reduction caused by an aggressive environment, instead of using a fracture parameter obtained from a test in air. For the experimental work, four medium and high strength steels exposed to a cathodic charge and cathodic protection environments were employed, studying two different loading rates in each case, and testing C(T) samples under slow rates in the environment. The study was completed with a subsequent fractographic analysis by SEM. A good degree of fulfilment was appreciated in both materials and environmental conditions, showing the validity of the predictions supplied by the FAD optimization model proposal, which constitutes an advance in the accuracy of the FAD predictive model.

**Keywords:** failure assessment diagram (FAD); hydrogen induced cracking (HIC); high strength low alloy steels (HSLA); cathodic protection (CP); cathodic polarization or cathodic charge (CC); subcritical propagation; micromechanisms; crack initiation

## 1. Introduction

The steels used in the power production industry, for instance in offshore and oil and gas facilities, have been evolving continuously over the last few decades. The main advances have taken place in improving their mechanical behavior in order to minimize the weight and optimize the transportation; their fracture toughness, in order to avoid problems caused by brittle fracture when working in aggressive environments and low temperatures; or their durability, in order to increase their resistance to subcritical cracking processes, among others [1].

Medium and high strength steels employed in the aforementioned fields are subjected to stress corrosion cracking (SCC) when working in marine environments involving cathodic protection, or sulfide induced cracking due to bacterial activity [2]. In both cases, hydrogen plays a fundamental role in the cracking mechanisms and therefore these steels should be resistant to hydrogen induced cracking (HIC). In these scenarios, the subcritical cracking mechanisms can be dominated by elastic-plastic

controlled critical conditions. Hence, the conventional characterization methods based on linear elastic-plastic mechanisms are not useful.

In the last few decades, new methodologies have been developed in order to characterize the SCC and HIC behavior of steels showing a high resistance to subcritical propagation processes [3,4]. These techniques are based on performing a lab test on pre-cracked specimens in the environment at slow rates, in order to obtain its load-crack opening displacement graph (F-COD), and from it parameters such as the toughness in environment or the subcritical crack growth rate [3]. In previous works carried out during the 2000s and 2010s [5–7], the methodology was completed by its direct application to structural design by means of failure assessment diagrams (FADs), a tool widely used in the evaluation of structural safety with highly satisfactory results. These diagrams provide a bi-parametric evaluation of structural integrity considering the combination of fracture and plastic collapse. The application of FAD to environmental processes is still a novel approach, only a few examples being found in the literature [5–7]. For this reason, the present work proposes improvements to the technique described in [5], which is based on a solid model, in order to simplify and universalize its application to HIC scenarios. The application of this analysis to HIC is a challenge of great interest that is not yet extended to standards.

However, as presented in [5], the possibility of subcritical propagation processes in HIC phenomena should be considered when applying FAD in structural design. For these situations, which do not lead to the fracture or collapse of the structure immediately, the subsequent cracked situations taking place during the propagation can be represented in the FAD diagram as a path from the crack initiation to the final failure of the component assessed [5].

In the aforementioned approach [5], the definition of the $K_r$ coordinate of the FAD is made by using the fracture parameter of the material in air (linear-elastic or elasto-plastic) [8,9], $K_{mat} \approx K_{Ic}$, $K_{mat} \approx K_{Jc}$ or $K_{mat} \approx K_{J0.2}$. However, it is well known that the fracture toughness in aggressive environments such as HIC, $K_{IHIC}$, is usually lower than its equivalent in air. Hence, a more brittle condition takes place [4,5,10,11], which will lead to higher $K_r$ values in the FAD representation of the subcritical crack path propagation [6].

In this work, a novel approach in order to optimize the application of FAD diagrams to the assessment of environmental assisted cracking propagation is proposed. It consists in introducing a simple correction in the methodology exposed in [5] for the definition of the $K_r$ coordinate of the FAD, allowing the fracture toughness reduction caused by HIC to be taken into account without the need to determine $K_{IHIC}$. It consists of obtaining an approximated $K_{mat}$ value in environment, $K_{mat-env}$, directly from the load-displacement (P-COD) curve obtained in the test by applying a proportionality between its maximum load and the one from a fracture toughness test in air on the same material.

In this work a new implementation of a crack propagation model over the FAD is also included, showing the capability and applicability of this tool in HIC processes.

All this is done with the aim of optimizing the FAD based prediction model, proposing an improved one whose validity is verified by testing four medium and high strength steels exposed to cathodic charge (CC) and cathodic protection (CP) environments, studying two different loading rates in each case.

The samples are tested by performing slow rate fracture mechanics tests on C(T) specimens, and the study is completed with a subsequent fractographic analysis by scanning electron microscope (SEM) techniques, in order to check the validity of the proposal from a micromechanical point of view.

## 2. Background

### 2.1. Behaviour in HIC Conditions

Fracture properties in HIC processes are commonly characterized by testing fracture mechanics specimens in the aggressive environment under study at a constant load or at a constant loading rate [12–14]. This last technique, constant loading rate, using C(T) specimens tested in a slow

rate machine is, probably, the most common, as it always leads to the specimen fracture, avoiding uncertainties. Hydrogen diffusion inside the material is promoted by exposure to the environment, normally during a certain amount of time prior to the mechanical solicitation [10–12] as well as during the test. A subsequent SEM fractographic study is often used to reveal the mechanisms behind the cracking process [10–12].

As explained in Figure 1, by applying the analytical methodology [4], based on the GE-EPRI procedure [3], it is possible to obtain crack size values throughout the test from the experimental P-COD curve, and also to determine the value of the J integral in their elastic and plastic components (and indirectly the stress intensity factor). As shown in Figure 1, it is possible to evaluate the crack propagation rate as a function of the stress intensity factor by using the concept of iso-a curves [3]. The method consists in their superposition to the P-COD curve from the test, and their intersection at the point where the crack length, *a*, has the same value as the iso-a curve crossing, as is explained in refernece [4] in detail.

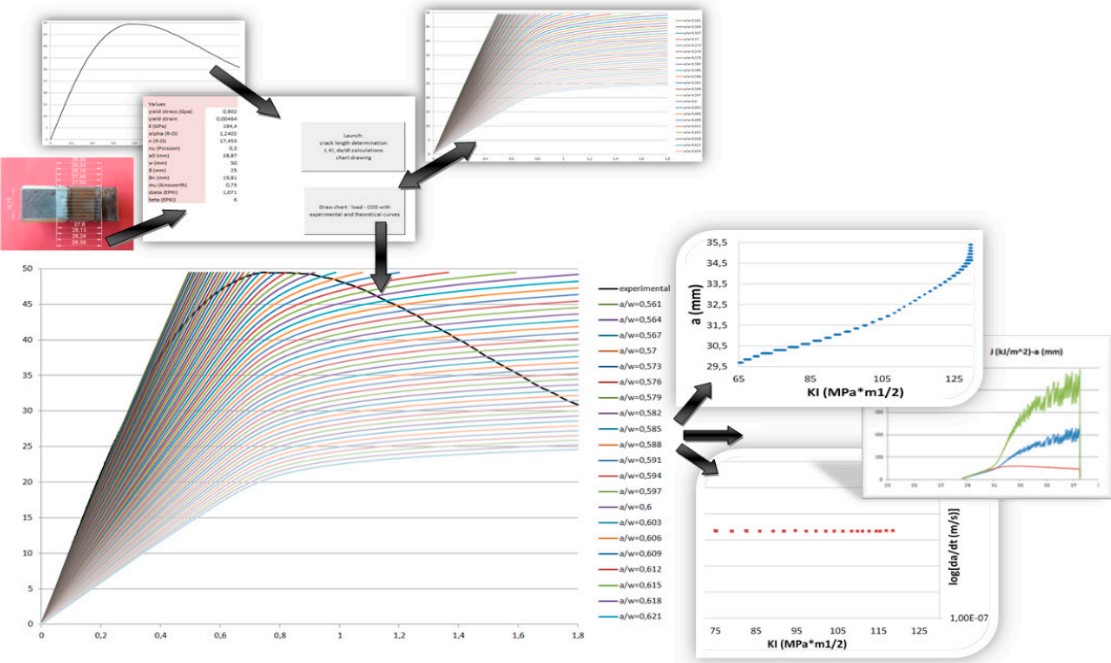

**Figure 1.** Example of the implementation of GE-EPRI methodology [3,4,15].

## 2.2. Crack Propagaion Model in HIC

General models attempt to explain the crack propagation modes on HIC conditions of high strength low allow steels (HSLA), mostly intergranular (IG) or transgranular (TG), and the parameters that control them, considering that the presence of hydrogen at the crack tip plays a fundamental role [4,5,16–19]. In [4,18], it is established that the crack propagation in HIC takes place as a series of local isolated fractures, nucleated and developed at the plastic zone ahead of the crack tip. These local fractures nucleate when the obtained plastic strain reaches a critical value determined by the embrittlement produced in this plastic zone by the presence of the hydrogen absorbed by the material, the solubility of the hydrogen being greater in this zone than in the rest of the metallic network. Thus, the model establishes that the crack propagation is controlled by the kinetics of the hydrogen throughout its entrance and diffusion through the crystal network.

The model exposed in [5] proposes that the nucleation of local fractures, which produces propagation, takes place in specific microstructure features or defects occurring either in the grain boundary or inside the grains, constituting traps in which the hydrogen concentration is very high and, consequently, the value of critical strain in them is particularly low.

The type of fracture that arises during propagation, IG or TG, is directly associated to the nucleation process described. If the nucleation occurs at the grain boundary, the fracture will be intergranular (IG), while if it occurs inside the grain it will be transgranular (TG). These different conditions are established through the competence between the grain size, $d$, the position of nucleation, $L^*$, and the size of plastic zone, $r_y$ (Figure 2). This, finally, leads to limiting expressions for the threshold values of IG or TG propagation as a function of the microstructure (grain size, $d$) and the mechanical behavior of the material (yield stress, $\sigma_y$, and Young modulus, $E$). These conditions transferred to macroscopic parameters, such as $K_I$, lead to the following expressions:

$$IG\ condition \rightarrow KIEAC < 0.85 \cdot \sqrt{\sigma_y \cdot E \cdot d}, \tag{1}$$

$$TG\ condition \rightarrow KIEAC > 0.42 \cdot \sqrt{\sigma_y \cdot E \cdot d}. \tag{2}$$

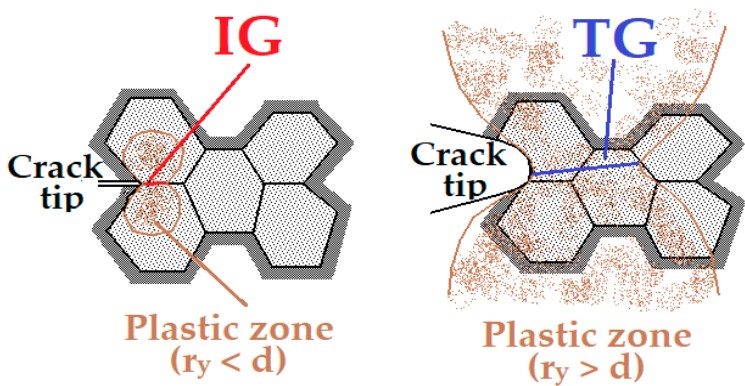

**Figure 2.** Explanation of intergranular (IG)/transgranular (TG) crack propagation as a function of the grain and the plastic zone sizes.

These conditions can be applied to P-COD representations for a given specimen as a function of crack length, in order to predict the transgranular or intergranular cracking zones. The extensive studies carried out in the area of hydrogen induced cracking [3] provide extensions of the model in order to cover the other micromechanisms observed at the crack propagation of medium and high strength steels. In [5], a new condition is established in order to differentiate the subcritical processes corresponding to low strain situations, as intergranular or pure cleavage. This takes place when it remains associated to local conditions controlled by values of the J integral dominated by its elastic component, $J_e$, from the cracking paths with tearing or microvoids coalescence associated to values of the J integral in which the plastic component, $J_p$, is significant.

This condition is defined by the following equation, corresponding in the P-COD representation to a constant displacement value independent of the crack length presented by the specimen [4,20].

$$J_e = 0.95 \cdot J. \tag{3}$$

According to the model from [5], Figure 3 shows the different defined zones in the P-COD plane for steels in HIC. The lines representing the conditions of interganularity (1) and transgranularity (2) define the IG, mixed IG/TG and pure TG (cleavage) zones. The boundary between elastic and plastic domains (3) and the plastic collapse condition for general yielding also define the tearing zone and the zone of plastic instability based on the nucleation growth and coalescence of microvoids.

In Figure 3, a set of P-COD curves that represent tests in one material under different environmental and loading rate conditions are represented. In all the cases, crack propagation is associated to the load decreasing line, which runs, depending on the condition, through different micromechanical zones.

The greater the aggressiveness in the testing procedure (higher current density, or voltage or slower displacement rate) the more brittle the conditions achieved.

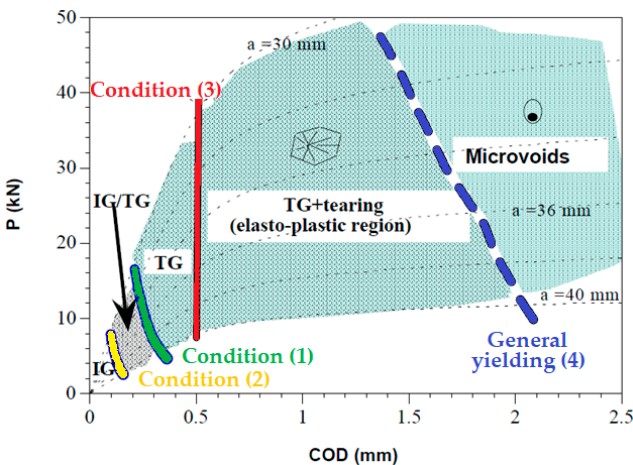

**Figure 3.** Modeling of the different behavior zones in the load displacement (P-COD) curve [5].

### 2.3. FAD Implementation of the Model

The approach based on failure assessment diagrams (FADs) establishes that failure in a structure or a component is avoided as long as the structure is not loaded beyond its maximum load capacity defined by using both fracture mechanics criteria and plastic collapse. The classical FAD is a plot of the failure envelope of the cracked component defined in terms of two parameters: $K_r$ and $L_r$, which are given by the expression below:

$$K_r = \frac{K_I}{K_{mat}}, \tag{4}$$

$$L_r = \frac{P}{P_y} \text{ or } L_r = \frac{\sigma_{ref}}{\sigma_y}, \tag{5}$$

where $K_r$ is the ratio of the applied linear elastic stress intensity factor, $K_I$, and the material fracture toughness, $K_{mat}$. $L_r$ is the ratio of the applied load, $P$, and the load that causes plastic yielding of the cracked structure, $P_y$; an alternative definition in terms of stresses can be given to $L_r$, being the ratio of the reference stress, $\sigma_{ref}$, and the yield stress, $\sigma_y$.

The failure envelope is called the failure assessment line (FAL) and is given by expressions proposed in the failure assessment procedures as in [8,20,21] as a function of the material's tensile properties; this line separates the safe and unsafe zones of behavior. Figure 4, obtained by using the software Vindio [22] that incorporates the expressions from [20,21], shows an example of the representation of a FAD diagram.

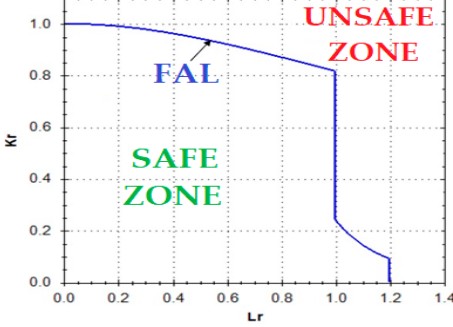

**Figure 4.** Failure assessment diagram (FAD) graphical representation.

The assessment of a component is based on the relative location of a geometry dependent assessment point respecting the failure line. The FAD diagram analysis allows us to determine the safety conditions without the need to determine if the component operates in small-scale yielding or plastic collapse regime. This is a great benefit, since this distinction would be complicated for a component. The application of this analysis to HIC is a challenge of great interest that is not yet extended to standards.

The previously presented model defines regions with common characteristics in the micromechanisms of cracking in the P-COD diagrams. The transfer from the P-COD representation, with defined zones for the micro-mechanical models, to a FAD diagram constitutes a further step in the unification of the design criteria based on local micro-mechanical approaches, which can be applied in any situation [5].

In [5], it is explained how the use of FAD can be amplified with the inclusion in the representation of the boundary conditions for IG (1) and TG (2) mechanisms in HIC processes, shown as functions of the stress intensity factor, corresponding to straight lines parallel to the horizontal axis (constant $K_r$) in the FAD representation. The third limit to define the HIC behavior is the initiation of the tearing zone (3), coincident with the development of the plastic component of the J integral. This line in the FAD-diagram corresponds to a representation parallel to the $K_r$ axis (constant $L_r$). The fourth limit, which defines the general plastification of the residual ligament (4), is coincident with the FAL line at the plastic collapse region, to the right of the failure line. Figure 5 shows the representation of a FAD including the different zones of propagation mechanisms defined by means of the corresponding limit conditions. In all cases, the function of these limits, and therefore the extension of the different zones, depends on the material.

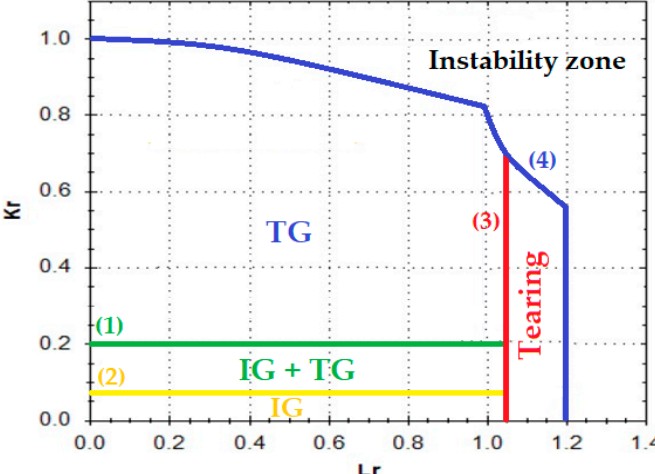

**Figure 5.** Zones for different hydrogen induced cracking (HIC) mechanisms in a FAD diagram in the original model.

## 3. Materials and Methods

### 3.1. Materials

Four materials, from two families, were used in this work:

- Mid-strength low alloy 2.25Cr-1Mo and 3Cr-1Mo-0.25V steels: Designed for working in harsh conditions, commonly used in offshore facilities and water treating reactors. In Figure 6, an image of their bainitic microstructure is presented, showing a relatively small grain size homogeneously distributed.
- High-strength 1Cr-1Ni-1.25Mn and 1Cr-1Ni-0.5Mo steels: Grades R5S and R6, respectively according to [23], obtained by quenching and tempering processes and then forged, employed in the manufacturing of offshore links of chains for mooring lines. In Figure 7, an image of its

martensitic and bainitic microstructure is presented, also showing a relatively small grain size homogeneously distributed.

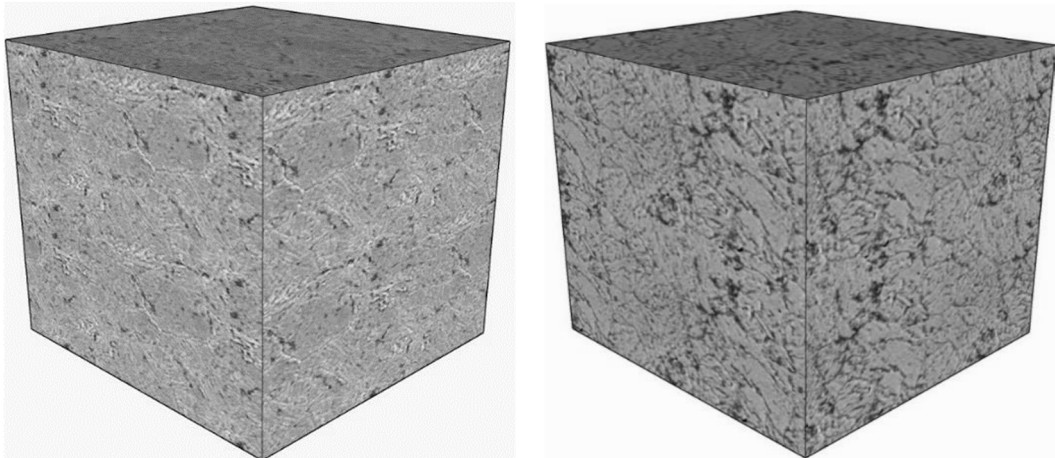

**Figure 6.** Microstructure of the 2.25Cr-1Mo (**left**) and 3Cr-1Mo-0.25V (**right**).

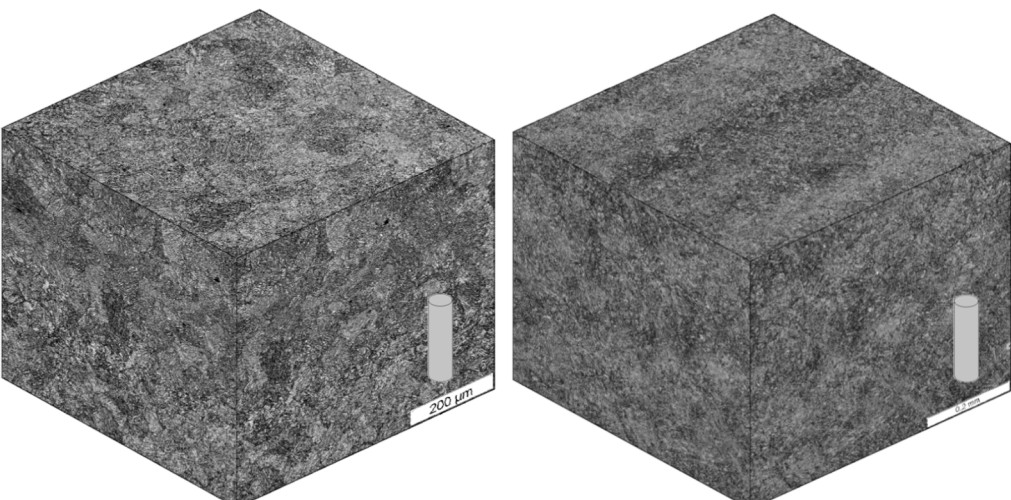

**Figure 7.** Microstructure of the 1Cr-1Ni-1.25Mn (**left**) and 1Cr-1Ni-0.5Mo (**right**).

In Table 1 the main tensile and fracture properties of all the steels used, as well as their grain size are presented.

**Table 1.** Main properties of the steels employed.

| Parameter | 2.25Cr-1Mo | 3Cr-1Mo-0.25V | 1Cr-1Ni-1.25Mn | 1Cr-1Ni-0.5Mo |
|---|---|---|---|---|
| $S_y$ (MPa) | 584 | 642 | 1024 | 1056 |
| $S_u$ (MPa) | 715 | 768 | 1120 | 1159 |
| $E$ (GPa) | 206 | 211 | 219 | 209 |
| $\varepsilon_{max}$ (%) | 7.3 | 5.8 | 6.0 | 6.2 |
| Ramberg Osgood $\alpha$ | 0.592 | 0.354 | 0.622 | 0.485 |
| Ramberg Osgood n | 9.4 | 18.5 | 17.0 | 18.0 |
| $J_{0.2}$ (KJ/m$^2$) | 662 | 241 | 821 | 760 |
| $K_{J0.2}$ (MPa·m$^{1/2}$) | 387 | 236 | 445 | 418 |
| Grain size (µm) | 49 | 17 | 48 | 47 |

*3.2. Environments Applied for Hydrogen Embrittlement*

On the one hand, the harsh environments in the oil and gas offshore industry involve the presence of hydrogen, which produces an embrittlement in the exposed materials. On the other hand, in the aforementioned applications, corrosion is always present; protection systems, such as cathodic protection (CP), are applied to minimize its effects, with the handicap that hydrogen embrittlement can also appear.

Hydrogen embrittlement (HE) will always have to be taken into account in these types of industrial facilities. In order to reproduce HE conditions, two different environmental conditions have been employed in this work. The first one, cathodic protection, is usually applied in the accessible parts of the platforms or off-shore structures. The second one, known as cathodic charge or anodic polarization, reproduces local aggressive environments that are impossible to avoid or predict that can seriously affect the structural integrity of the component exposed.

- Cathodic protection (CP) is used to avoid the corrosion phenomena in marine water environments. It involves the use of a sacrificial anode of aluminum (more active than steel), which, in the presence of seawater, is connected to the steel structure. This is the cathode that will be protected from corrosion [24] due to the imposition of a fixed potential between the two, which will maintain the stability of the process. In this study, an aggressive environment of marine water was simulated, consisting of a 3.5% in weight dissolution of NaCl in distilled $H_2O$ [11]. An aluminum anode was employed. The pH was controlled at the range 5.5–5.7 [11] during the whole duration of the tests at room temperature 20–25 °C. The level of cathodic protection (aggressiveness) employed was 1050 mV of fixed potential imposed.

- Cathodic charge (CC), or cathodic polarization, is used to reproduce situations where a huge amount of hydrogen is present, such as acid environments or local situations of hydrogen concentration. It consists in the interconnection, via an acid electrolyte, of a noble material (platinum in this case) and the steel, which will passivate and receive protection due to the fixed current interposed between the two [24]. In this work, the levels of current interposed were of 5 mA/cm$^2$ (of submerged sample). The aqueous environment employed was an acid electrolyte consisting of a 1N $H_2SO_4$ solution in distilled $H_2O$, with 10 mg of an $As_2O_3$ solution and 10 drops of $CS_2$ per liter of dissolution. The $As_2O_3$ solution was prepared following the Pressouyre's method [4,10,25]. The pH was measured in the range 0.65–0.80 during the tests and at room temperature (20–25 °C). The resulting protection causes the hydrogen atoms to be absorbed on the host lattice [26,27], producing local or global embrittlement.

A schema of the cathodic protection (CP) and cathodic polarization (CC) set-ups employed during the tests is shown in Figure 8. In both cases, the aqueous solution is in continuous circulation and/or agitation, as recommended by [12], in order to remove hydrogen bubbles on the specimen surface, avoiding deposits or local environmental conditions.

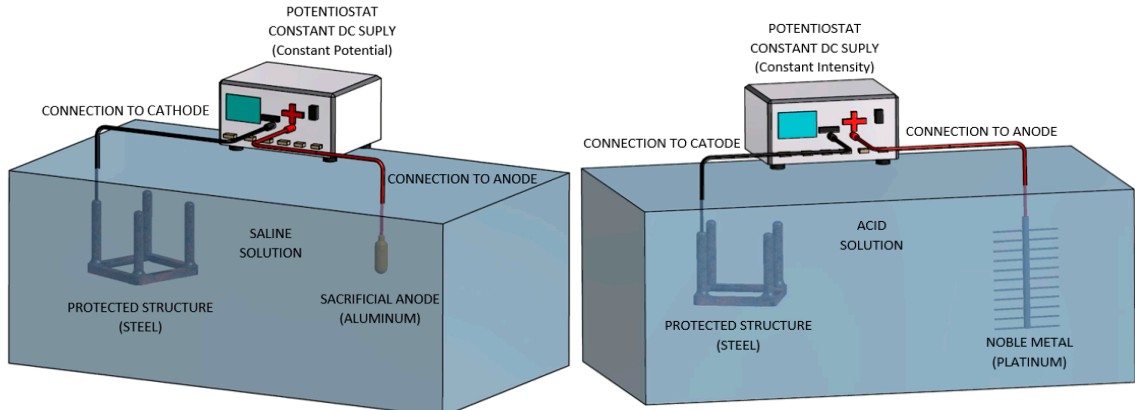

**Figure 8.** Schema of cathodic protection (left) and cathodic polarization (right) set-ups.

*3.3. Optimization Proposals for the Model Presented in Epigraph 2*

In the present work, an optimization of the model exposed in [5] (epigraph 2) is presented. It consists of the improvement of the following aspects:

- Slight modification of the zones in which the FAD is divided for HIC conditions. As previously outlined in Section 2 and shown in Figure 5, the FAD diagram is divided into different zones for the different HIC mechanisms taking place (IG, IG/TG, TG, Tearing). On most occasions, depending on the material and the environment, a pure IG zone is very limited; in practice there is a mixed IG/TG mode from the beginning of the propagation. For this reason, in this work, a modification to merge the zones of pure IG and IG + TG into a single one is proposed, modifying the FAD model zones to the one presented in Figure 9 (comparison of Figures 5 and 9).

- Definition of the crack propagation initiation in the elastic-plastic range. For this purpose, the concept of the iso-*a* slope in its straight initial part is employed. Crack initiation is marked at the point in which the iso-$a_{\text{ini}}$ curve intersects the P-COD register, the iso-$a_{\text{in}}$ being the one that has a slope at the initial straight part equal to 95% of the iso-$a_0$ (iso-*a* corresponding to the crack length prior to the test, $a_0$). By using this approach, it is possible to define the crack initiation in an elastic-plastic field, rather than in the linear elastic fracture mechanics (LEFM) one that, for example, ASTM standards [9] recommend. In Figure 10, a graphical representation of the new approach, as well as the LEFM one, is given.

- The last proposed correction is focused on the crack propagation path representation in the FAD. It consists in introducing a modification in the definition of the $K_{\text{r}}$ coordinate ($K_{\text{r}} = K_{\text{I}}/K_{\text{mat}}$) in each of the points that define the crack path propagation in the FAD. It allows the fracture toughness reduction caused by an aggressive environment to be taken into account, without the need to determine $K_{\text{IHIC}}$. The FAD application to environmentally assisted processes leads to predictions far from real behavior, due to the fact that it applies for its calculation a fracture toughness value obtained in non-aggressive scenarios. Applying the aforementioned model, described in Section 2.2, on FAD diagrams and superposing crack path propagations, which is a current trend for predicting propagation and failure in high-plasticity conditions, gave results that did not match with real behavior, as shown in Figure 11. The proposed correction (Figure 11) consists in obtaining an approximate $K_{\text{mat}}$ value in environment, $K_{\text{mat-env}}$, directly from the load-displacement (P-COD) curve obtained in the test. This is done by applying a proportionality between the values of the test in environment and their equivalent values from a fracture toughness test in air, performed on the same material. This correction can be expressed as follows, $K_{\text{r}}$ then being:

$$K_{\text{mat-env}} = K_{\text{mat}} \cdot \frac{P_{\text{max-env}}}{P_{\text{max}}}, \tag{6}$$

$$K_{\mathrm{r}} = \frac{K_{\mathrm{I}}}{K_{\mathrm{mat-env}}}. \tag{7}$$

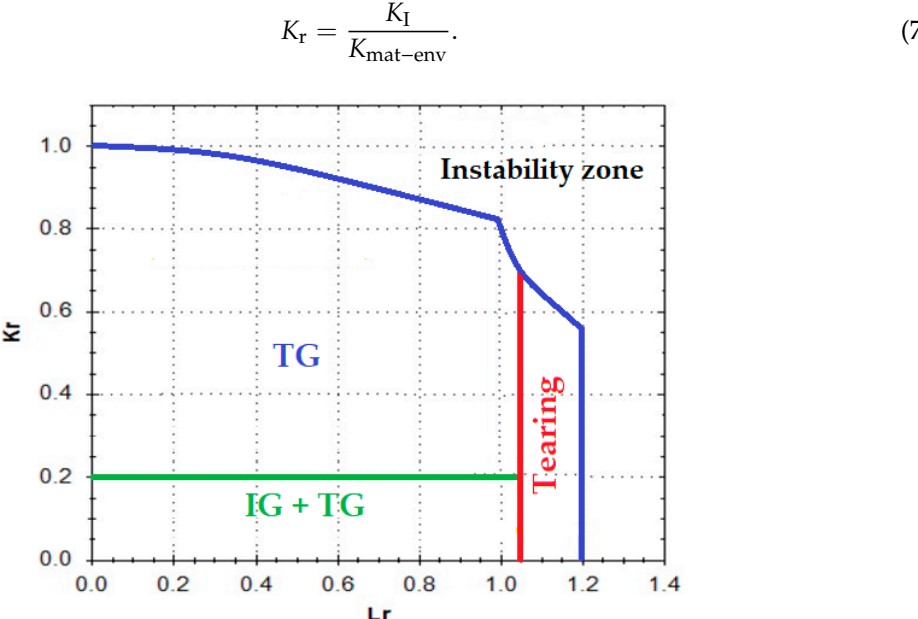

**Figure 9.** Proposal for zones for different HIC mechanisms in a FAD diagram merging IG with IG + TG.

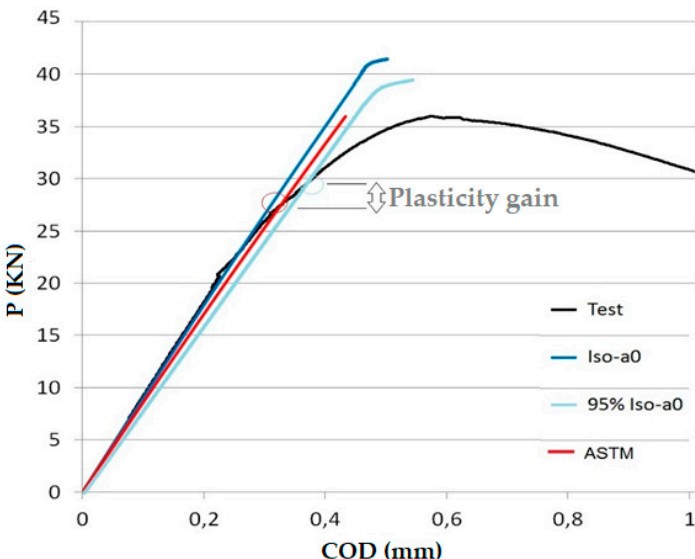

**Figure 10.** Proposal for the definition of the crack propagation initiation based on the elastic-plastic range.

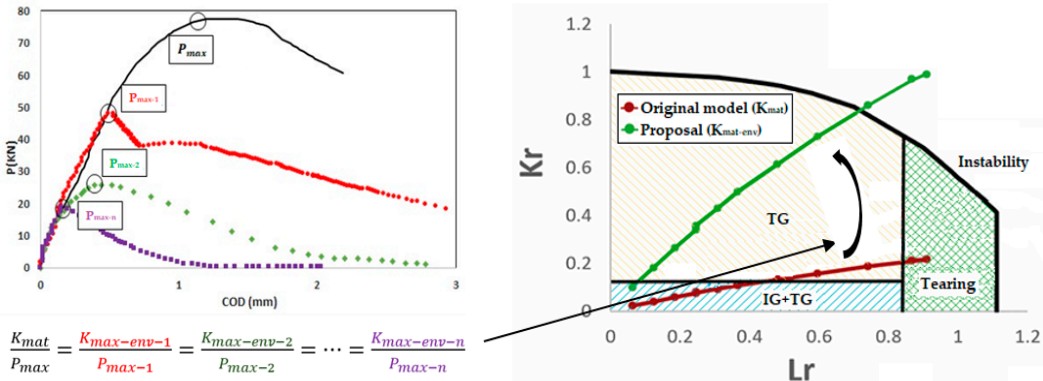

$$\frac{K_{mat}}{P_{max}} = \frac{K_{max-env-1}}{P_{max-1}} = \frac{K_{max-env-2}}{P_{max-2}} = \cdots = \frac{K_{max-env-n}}{P_{max-n}}$$

**Figure 11.** Proposal for the crack propagation crack representation on the FAD.

By applying this modification, it is possible to correct those situations where the embrittlement is substantial. $K_{mat-env}$ is appreciably lower than $K_{mat}$ so using $K_{mat}$ is on the unsafe side, although it continues to be accurate when the embrittlement is not very great. $K_{mat-env}$ is close to $K_{mat}$. As shown in Figure 11, the result from its application will lead to more brittle crack paths in the FAD; $L_r$ values will remain the same while $K_r$ ones will be higher ($K_{mat-env}$ is lower than $K_{mat}$), which means that the crack path drawn in the FAD will have a higher slope ($K_r/L_r$).

*3.4. Test Plan*

HIC processes were characterized by testing 25 mm thick C(T) specimens according to [9,12,28]. The samples were pre-cracked by $R$ = 0.1 fatigue load according to [28], and then 20% side-grooved. After that, the P-COD slow strain tests were performed in the aforementioned environments on the steels selected (epigraph 3.2). In order to reproduce industrial situations, the acid cathodic charge environment (CC) was applied on the mid-strength low alloy 2.25Cr-1Mo and 3Cr-1Mo-0.25V steels, under a current constant intensity, $i$, of 5 mA/cm$^2$; the rate effect was studied by testing at displacement rate, $V_d$, from $10^{-7}$ to $10^{-9}$ m/s in each of the possible situations. In order to reproduce off-shore environments, the cathodic protection environment (CP) was also applied on the high-strength 1Cr-1Ni-1.25Mn and 1Cr-1Ni-0.5Mo steels, imposing a constant potential, E, of 1050 mV; in this case rates, $V_d$, from $6.10^{-8}$ to $6.10^{-9}$ m/s were analyzed in each situation. By performing the described combinations of material-environment-rate, a wide range of microstructures and different hydrogen embrittlement scenarios were covered, as shown in Table 2 where the plan tests are presented.

**Table 2.** Test plan carried out.

| Material | Environment ($i$,E) | | Rate ($V_d$) |
|---|---|---|---|
| 2.25Cr-1Mo | Cathodic Charge | 5 mA/cm$^2$ | $10^{-7}$ m/s $10^{-9}$ m/s |
| 3Cr-1Mo-0.25V | Cathodic Charge | 5 mA/cm$^2$ | $10^{-7}$ m/s $10^{-9}$ m/s |
| 1Cr-1Ni-1.25Mn | Cathodic Protection | 1050 mV | $6 \times 10^{-8}$ m/s $6 \times 10^{-9}$ m/s |
| 1Cr-1Ni-0.5Mo | Cathodic Protection | 1050 mV | $6 \times 10^{-8}$ m/s $6 \times 10^{-9}$ m/s |

Prior to the test, the samples were exposed to the corresponding liquid aggressive environment for 48 h (1 N $H_2SO_4$ solution in $H_2O$ under 5 mA/cm$^2$ for cathodic charge, or simulated marine water under 1050 mV for cathodic protection), in order to polarize them. After that, the mechanical load was applied maintaining the sample exposed to the environment during the whole test. The mechanical testing consisted in the application of a constant loading rate using a slow rate tensile machine of horizontal axis; crack propagation during testing led to the specimen's rupture. In Figure 12, a set-up of the test is presented. After the test, the samples were cleaned using ultrasounds while submerged in acetone and dried. A later SEM fractographic study revealed the mechanism behind the cracking process.

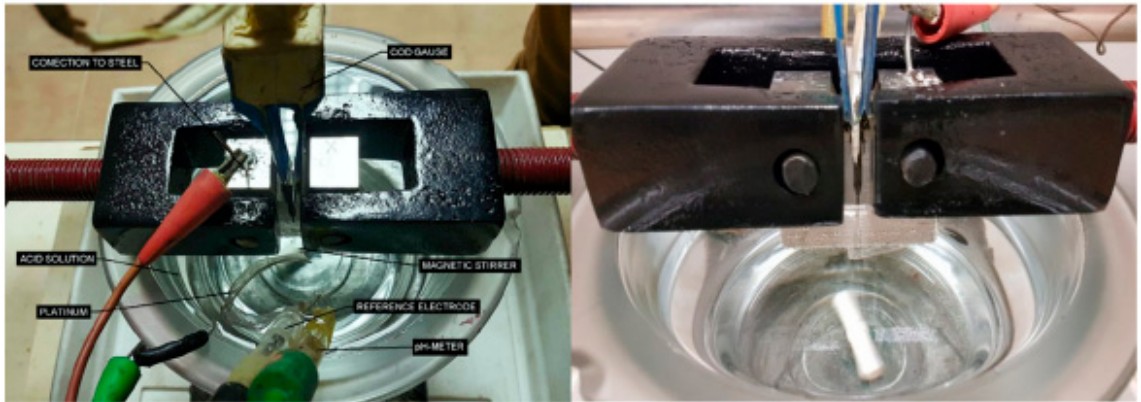

**Figure 12.** Set-up of the slow rate tests in environment using C(T) specimens.

## 4. Results

In Figure 13, the P-COD registers from the slow rates tests performed are presented. In Figures 14–17, the subsequent SEM fractographic analysis shows the microstructure during the propagation, positioning the images at their corresponding point of the P-COD register.

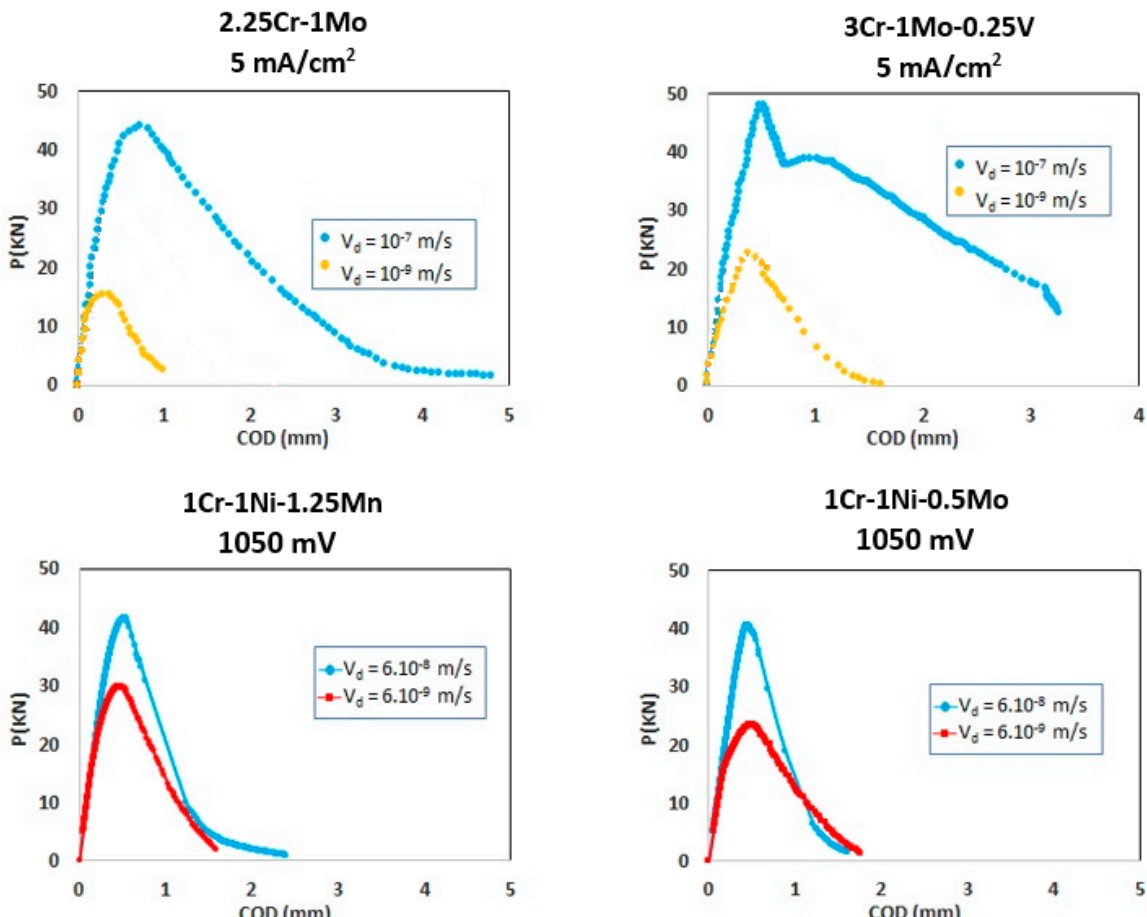

**Figure 13.** P-COD registers from the slow rate tests in the environment.

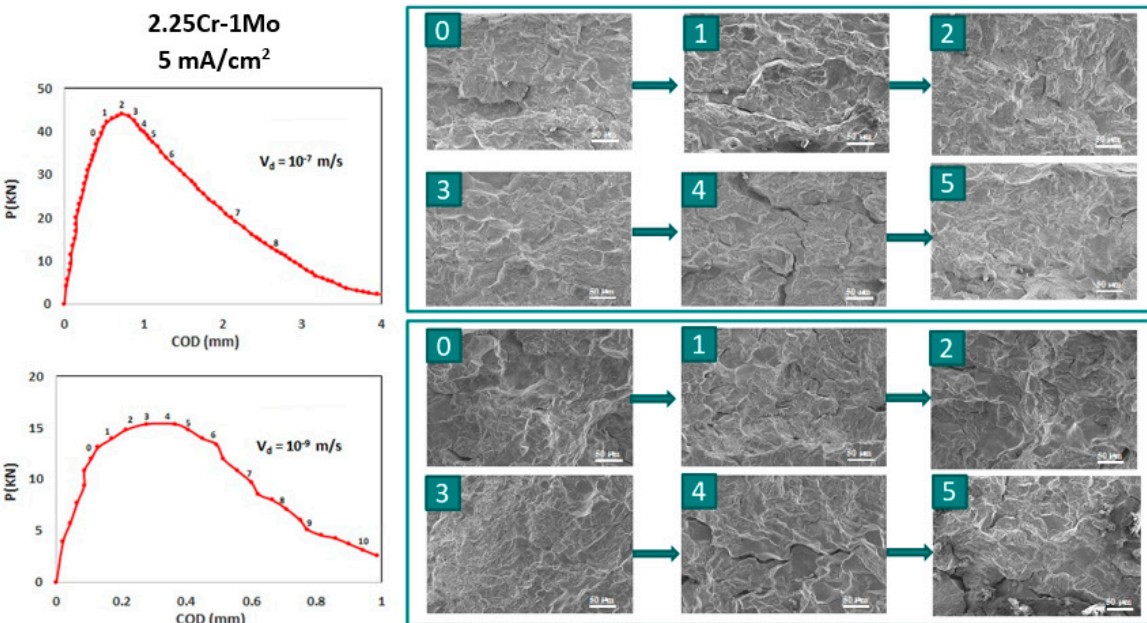

**Figure 14.** Fractographic images corresponding to different stages of P-COD register from the slow rate tests performed on 2.25Cr-1Mo steel under 5 mA/cm$^2$ CC environment at $10^{-7}$ and $10^{-9}$ m/s.

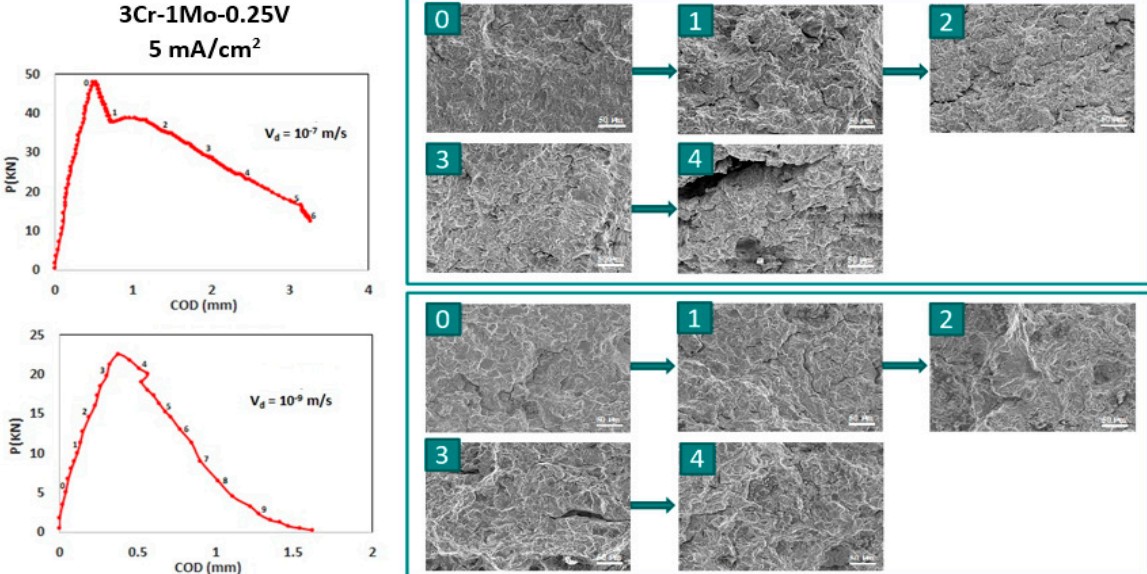

**Figure 15.** Fractographic images corresponding to different stages of P-COD register from the slow rate tests performed on 3Cr-1Mo-0.25V steel under 5 mA/cm$^2$ CC environment at $10^{-7}$ and $10^{-9}$ m/s.

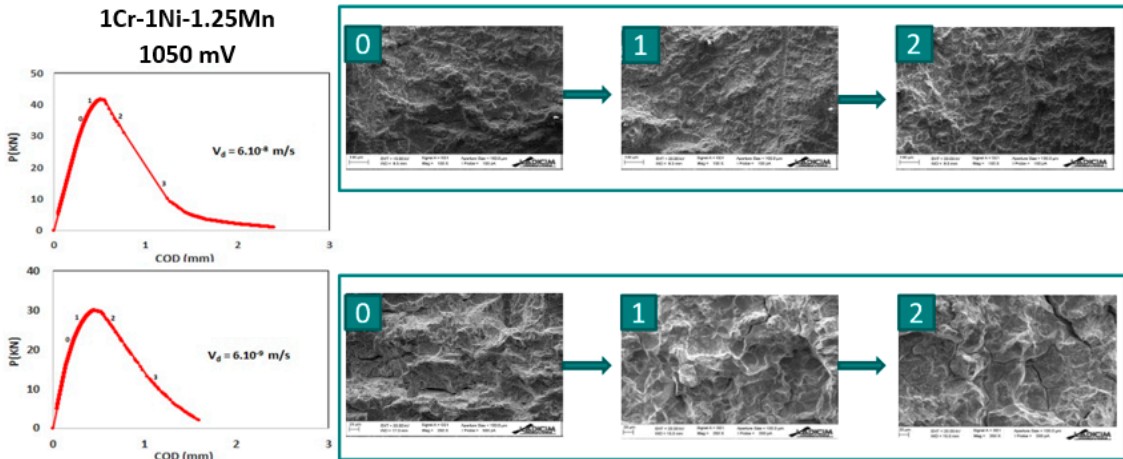

**Figure 16.** Fractographic images corresponding to different stages of P-COD register from the slow rate tests performed on 1Cr-1Ni-0.25Mn steel under 1050 mV CP environment at $6 \times 10^{-8}$ and $6 \times 10^{-9}$ m/s.

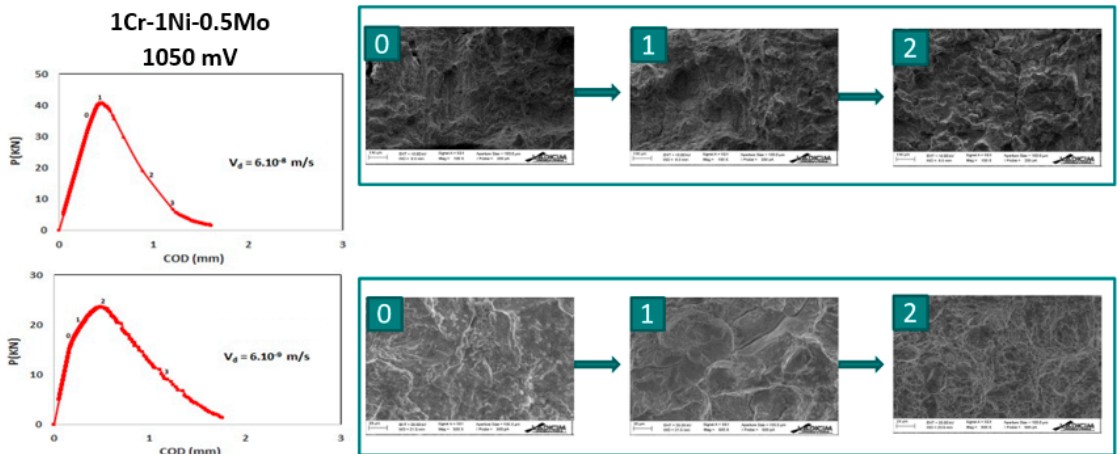

**Figure 17.** Fractographic images corresponding to different stages of P-COD register from the slow rate tests performed on 1Cr-1Ni-0.5Mo steel under 1050 mV CP environment at $6 \times 10^{-8}$ and $6 \times 10^{-9}$ m/s.

From the previous results, in Figures 18–21, the FAD diagrams, including the different zones in which they are for HIC conditions (see Figure 9) and the optimization proposals to the model (epigraph 3.3), are presented. In each case, in addition to the crack propagation path across the different zones, selected SEM micrographs are presented at certain points, in order to be able to verify whether the microstructure taking place along the crack propagation satisfies the proposed modified model.

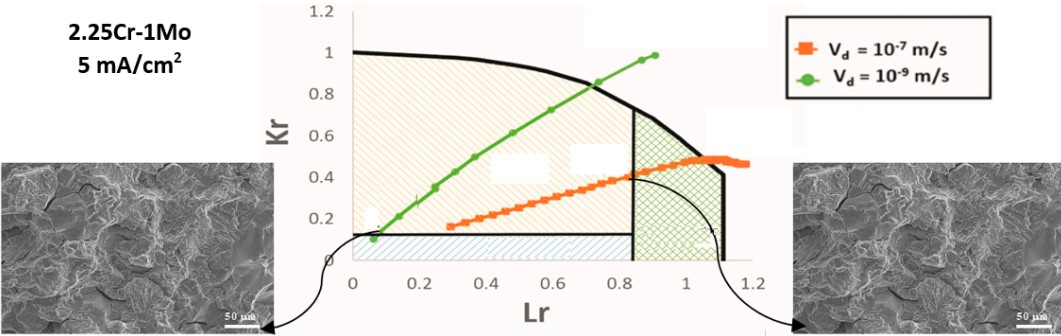

**Figure 18.** Crack path propagation displayed on the FAD, and fractography, from the slow rate tests performed on 2.25Cr-1Mo steel under 5 mA/cm$^2$ CC environment at $10^{-7}$ and $10^{-9}$ m/s.

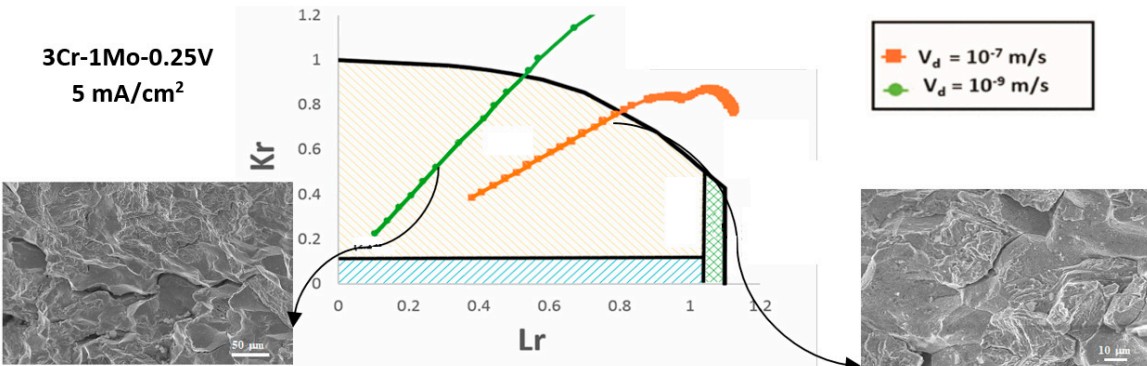

**Figure 19.** Crack path propagation displayed on the FAD, and fractography, from the slow rate tests performed on 3Cr-1Mo-0.25V steel under 5 mA/cm$^2$ CC environment at $10^{-7}$ and $10^{-9}$ m/s.

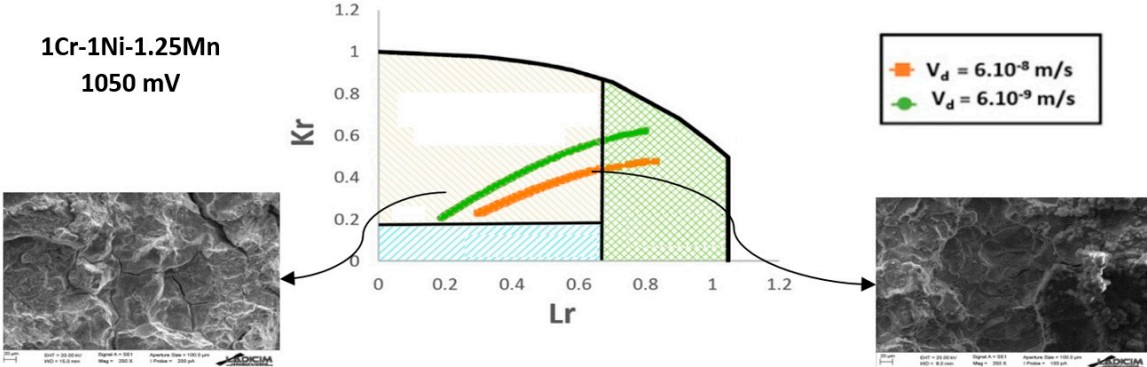

**Figure 20.** Crack path propagation displayed on the FAD, and fractography, from the slow rate tests performed on 1Cr-1Ni-0.25Mn steel under 1050 mV CP environment at $6 \times 10^{-8}$ and $6 \times 10^{-9}$ m/s.

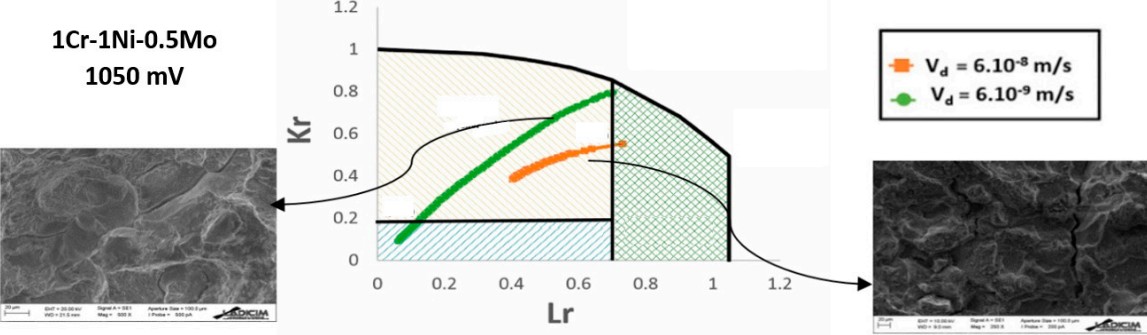

**Figure 21.** Crack path propagation displayed on the FAD, and fractography, from the slow rate tests performed on 1Cr-1Ni-0.5Mo steel under 1050 mV CP environment at $6 \times 10^{-8}$ and $6 \times 10^{-9}$ m/s.

## 5. Discussion

By examining the P-COD registers, presented in Figure 13, and the SEM fractographic analysis, presented in Figures 14–17, it can be stated that the steels employed in this work have shown susceptibility to the HIC processes used, following the classic rule of dependence on the loading rate. As the rate decreases, the material susceptibility, in a given environmental condition, increases and more brittle micromechanisms are present because the hydrogen available at the crack tip is higher.

By examining Figures 18–21, showing the crack propagation paths across the different zones of the FAD diagrams, it can be stated that:

- 2.25Cr-1Mo steel:

    - $10^{-7}$ m/s: The initial subcritical process started in the cleavage domain (pure TG, but close to the mixed IG + TG), and the subsequent propagation ran under the cleavage domain (TG) and the plastic domain (Tearing) prior to crossing the failure line (FAL) to the instability zone.
    - $10^{-9}$ m/s: The process started at the mixed cleavages domain (IG + TG, but very close to the pure TG), and propagated across the TG zone until crossing FAL to the instability zone. No tearing was shown.

- 3Cr-1Mo-0.25V steel:

    - $10^{-7}$ m/s: The process started at the mixed cleavages domain (IG + TG) and propagated across the TG zone until crossing FAL to the instability zone. No tearing was shown.
    - $10^{-9}$ m/s: The propagation started at the cleavages domain (IG + TG), and followed across the TG zone until crossing the FAL. No tearing was shown.

- 1Cr-1Ni-1.25Mn steel:

    - $6 \times 10^{-8}$ m/s: The process started at the TG domain (but close to the mixed IG + TG) and propagated across it up to the plastic domain (Tearing), where the test was stopped (it would have crossed the FAL to the instability zone if continued).
    - $6 \times 10^{-9}$ m/s: The process started at the TG zone (but very close to the mixed IG + TG) and propagated across the TG zone up to the plastic domain (Tearing), where the test was stopped (it would have crossed the FAL to the instability zone if continued).

- 1Cr-1Ni-0.5Mo steel:

    - $6 \times 10^{-8}$ m/s: The process started at the TG domain and propagated across it up to the plastic domain (Tearing), where the test was stopped.
    - $6 \times 10^{-9}$ m/s: The process started at the mixed cleavages domain (IG + TG) and propagated across the TG zone up to the plastic domain (Tearing) very close to the triple point of coincidence of TG-Tearing-FAL (see the green curve on Figure 21), where the test was stopped (it would have crossed the FAL to the instability zone if continued).

In the four combinations of material-environment (two different rates in each case), the rate effect was verified, as previously stated from the analysis of the P-COD curves. In all cases, the rate lowering had an embrittling effect, leading to more brittle path propagation curves in the FAD representation, and thus paths that had a higher slope.

The subcritical propagation grew across the cleavage (TG) region for the most aggressive environments, those with a cathodic charge (CC), until crossing the FAL to the unsafe zone of the FAD. In the 2.25Cr-1Mo and 3Cr-1Mo-0.25V steels under CC, only the least aggressive situation tested (2.25Cr-1Mo at $10^{-7}$ m/s) invaded the tearing region; these predictions were in accordance with the SEM images obtained.

In the case of the least aggressive environment, cathodic protection (CP), the propagation grew across the cleavage (TG) region towards the tearing region. In the 1Cr-1Ni-1.25Mn and 1Cr-1Ni-0.5Mo steels under CP, only the most aggressive situation tested (1Cr-1Ni-0.5Mo at $6 \times 10^{-9}$ m/s) invaded the mixed IG + TG region at the crack initiation, and it was also very close to the triple point of coincidence of TG-Tearing-FAL (green curve on Figure 21) in the tearing zone. These predictions were in accordance with the SEM images obtained.

It can be noticed that the extension of the different zones (IG + TG, TG cleavage or tearing) is different for both materials, but the obtained SEM fractographs showed the validity of the predictions supplied by the model. A good degree of fulfilment could be appreciated in both materials and

environmental conditions between the cracking processes and predictions, using the optimization proposal described here. Basically, for the materials, environments and rates studied, the model predicts a crack propagation path in the cleavage TG region for the most aggressive environments, up to the FAL line, while for systems less affected by HIC, the crack starts propagating in the mixed IG + TG zone to pass across the cleavage TG zone and develop tearing before crossing FAL.

## 6. Conclusions and Future Work

In this work, a novel approach in order to optimize the application of FAD diagrams to the assessment of environmental assisted cracking propagation is proposed. In order to do this, firstly, a general review of medium and high strength steel behavior in HIC conditions and the main crack propagation characteristics in these environments was undertaken. Then, the original model for crack propagation regions based on the micromechanical comparison by SEM images and its application to failure assessment diagrams (FAD), presented in [5], has been described in detail, in order to be able to optimize it.

The optimization proposed consisted of three approaches. The first one was the definition of the crack propagation initiation in the elastic-plastic range by using the iso-*a* slope in its straight initial part. The second one was a slight modification of the zones in which the FAD was divided for HIC conditions, merging pure IG (which can be very limited on some occasions) and the mixed zone of IG + TG into a single region to make it more practical for its application. The third and main approach consisted of introducing a simple correction for the definition of the $K_r$ coordinate of the FAD to take into account the fracture toughness reduction caused by an aggressive environment without the need to determine $K_{IHIC}$. This consisted of obtaining an approximate $K_{mat}$ value in the environment, $K_{mat-env}$, directly from the load-displacement (P-COD) curve obtained in the test by applying a proportionality between its maximum load and the one taken from a fracture toughness test in air on the same material.

For the experimental work, four medium and high strength steels exposed to cathodic charge (CC) and cathodic protection (CP) environments, studying two different loading rates in each case, have been employed. The samples were tested by performing slow rate fracture mechanics tests on C(T) specimens, and the study was completed with a subsequent fractographic analysis by SEM techniques, in order to verify the validity of the proposals from a micromechanical point of view.

In all the combinations of material-environment-rate, the rate effect was observed; the rate lowering had an embrittling effect, leading to more brittle path propagation curves in the FAD representation. This susceptibility was in compliance with the rules of classic HIC dependence. The obtained SEM fractographs showed the validity of the predictions supplied by the FAD optimization model proposal; a good degree of fulfilment could be appreciated in both materials and environmental conditions.

It was proved that both P-COD and FAD approaches show their capacity to be a useful tool in order to relate the micromechanisms responsible for crack propagation once the different zones have been defined in them. Nevertheless, the FAD, as a tool independent of the component geometry, has proven to be very useful for the assessment of possible HIC processes and their effect on structural integrity. For this reason, the optimization proposed constitutes an advance in the accuracy of the FAD predictive model.

The future work in this field should basically focus on three aspects. First, on the extension of the optimizations proposed to a wider range of materials and environments, and to other specimen geometries (as the FAD is non-dependent of the geometry); second in making their predictions more accurate by incorporating a more precise definition of the $K_r$ coordinate of the crack propagation; finally, as shown in Figure 22 as a proposal for possible future work, a finer micromechanical analysis could help to obtain an improvement in the definition of the correction proposed for $K_{mat}$ ($K_{mat-env}$). Although this involves taking the original model a step further, it consists of a simple proportionally that can be tuned to be more precise.

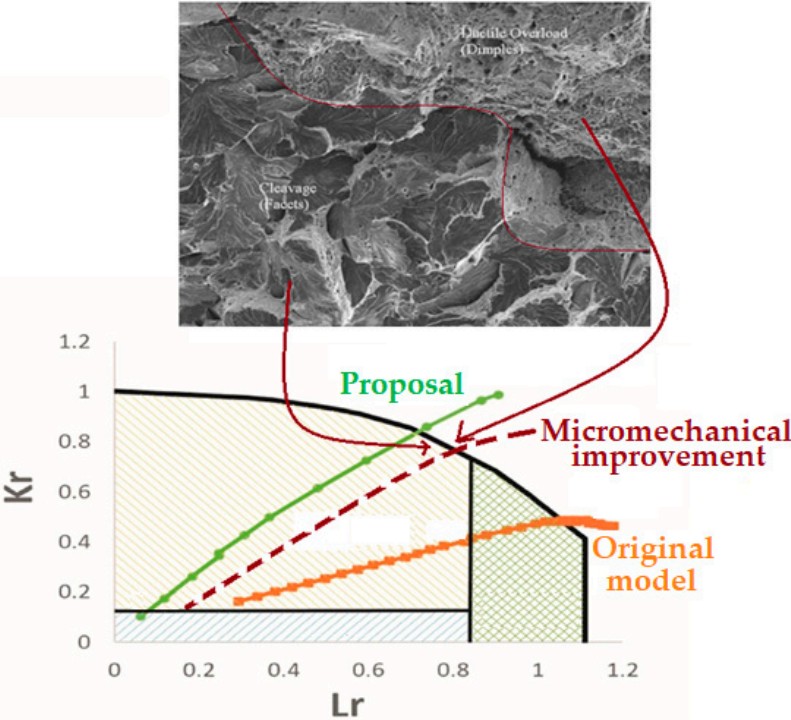

**Figure 22.** Improvement proposal for possible future work based on the work presented.

**Author Contributions:** Conceptualization, B.A.M., J.A.Á.L. & F.G.-S.; methodology, B.A.M., J.A.Á.L. & F.G.-S.; validation, B.A.M., A.C.M. & Y.J.J.M.; formal analysis, B.A.M., J.A.Á.L. & F.G.-S.; data curation, B.A.M., A.C.M., Y.J.J.M., & A.R.S.A.; FAD representation, A.R.S.A.; writing—original draft preparation, B.A.M. & J.A.Á.L.; writing—review and editing, B.A.M. & J.A.Á.L.; supervision, J.A.Á.L. & F.G.-S.; project administration, J.A.Á.L. & F.G.-S.; funding acquisition, J.A.Á.L. & F.G.-S.

**Funding:** This research was funded by CECA, grant number 7210-PE/110, by The Spanish Ministry of Economy and Competitivity, grant number MAT2014-58738-C3-3-R, and by the post-doctoral contracts program of the University of Cantabria, budgetary application 62.0000.64251.

**Conflicts of Interest:** The authors declare no conflict of interest.

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
