# Peer review of "A Proposal for the Application of Failure Assessment Diagrams to Subcritical Hydrogen Induced Cracking Propagation Processes"

_metals, doi:10.3390/met9060670_

Round 1
Reviewer 1 Report
Authors presented a research dealing with optimization of well known FAD diagram when it is applied to the assesment of environmetally influenced cracking of steel. It is a matter of relative interest to the audiance, given that FAD is more commonly used in real-life structures. Paper could be recommended for publishing in Metals, but the manuscript must undergo serious update, as follows:
List of references in the text start with 10, 11, 12... instead of 1, 2, 3...
Paper is founded on a reference [13], a conference paper old almost 20 years. Atuhors should provide better motivation for their work.
List of references relies heavily on rather old papers, industrial standards and PhD thesis. It should be updated with recent work on the topic to show relevance.
Reference for statement in 55-57 is missing.
Line 73: "sewing"?
Throughout the text there is "Error! Reference source not found." warning displayed making it impossible to link text and appropriate reference.
Fig. 1 - what is the purpose of it?
Throughout the manuscript, figures are not mentioned in the text.
Figures 5 and 9 are the same.
Define "liquid agressive environment" in line 329.
Author Response
Dear reviewer,
Thank you very much for your comments in order to improve our work. All of them were addressed as follows:
- The list of references was accidentally modifed during the post-processing. It was corrected. A similar thing happened with the list of tables and figures appearing “error, reference not found”; this was also fixed.
- A research in literature about aggressive envrionments and FAD diagrams was carried out, finding many contriutions in both topics separatelly. However, the application of FAD in HIC environments (combination of both topics) is still novel. Some works, in which the paper is founded (reference [5]), were carried out by the authors of the present paper in the early 2000’s, Some other works published in the 2000’s and 2010’s have been found and referenced. Based on this an optimization of our prior research has been done, incorporating actual materials and commonly employed embrittlement methods.
- References for the statements in lines 55-57 were included.
- “Sewing” was a mistake, it was meant “scanning”; this was corrected.
- The intention of Figure 1 is to explain the GE-EPRI methodology that can be employed to evaluate the crack lenght (crack propagation) from the P-COD register. It was indicated in the text that this is the purpose.
- Figure 5 belongs to the original FAD based model form reference [5], while Figure 9 is one of the proposals included in present work. Notes in Figure 5 caption and in the part of the text prior to Figure 9 was included in order to clarify this.
- “Liquid aggressive environment” means 1N H2SO4 solution in H2O under 5 mA/cm2 for cathodic charge, or simulated marine water under 1050 mV for cathodic protection. A note in the text was included to clarify this.
Reviewer 2 Report
I paper deals with very interesting and useful subject, however the quality of Engligh is very poor. Some sentences are too long and not understandable.
For example in the title the word Optimization is likely not the correct one. Also, in the title no abbreviations are accepted.
One important comment out of the English: You are writting about EAC in general, but in fact all you examples and explanations are about HIC. I do not agree that HIC belongs to EAC. In my understanding EAC is used to decribe effects of water or vapour environments. It includes the effect of environment on the crack growth in the moment when it grows. In contrary, HIC as you decribes in the paper (lines 96-106) consists of (i) diffusion and accumulation of hydrogen in front of the crack tip and (ii) brittle fracture of the region. I would propose that you change EAC to HIC in the title and further in text.
Moreover, look at your reference numbers used in the text. In introduction, the numebring starts from 10, but it should start from 1. Also, many references are not showing and the message Error appears.
Figure 22: Does the improvement proposal means the optimization you wrote about?
Author Response
Dear reviewer:
Thank you very much for your comments in order to improve our work. All of them were addressed as follows:
- English quality was improved as much as it was posible. But if it is still not enought it can be send to a native English speaker in order to check it in a second revision round (as the due date was in 5 days this couldn’t be done).
- Thank you for your coment, the title was modified in order to eliminate abreviations and focus it to the content treated.
- Your comments about not including HIC inside EAC processes were very clarifiying. In order to be rigorous all terms EAC were replaced by HIC, enven including it in the title.
- Figure 22 corresponds to a proposal for possible future work. This was clarified in the text as well as in the figure caption.
- The list of references was accidentally modifed during the post-processing. It was corrected. A similar thing happened with the list of tables and figures appearing “error, reference not found”; this was also fixed.
Reviewer 3 Report
The manuscript is dealing with fatigue crack growth in two different alloys considering hydrogen embrittlement and proposing a procedure to estimate the fatigue life. The structure of the manuscript is confusing and not well written and in some cases, it is difficult to understand and follow. The abstract is not presenting the contribution to knowledge and the novelty. The terminology adopted is not appropriate such as “hydrogen entry”, “breakage”, “expression collected”, “non-unstable propagation”. The fracture toughness is conventionally indicated with the symbol KIC and not Kmat as you did. Line 85 “suitable methodology GE-EPRI” not explained. The use of iso-a should be described. The references in the paper are starting from 10 and there is a “reference source not found” repeated several time in the paper.
The SEM analysis, which is shown in figures 13-16, it is not used to support the different mechanisms (IG, TG and tearing). The pictures seem to be reported wihout any clear justification. Paragraph from line 259 to 265 is introducing an optimisation, which uses a model “previously exposed”, but it is difficult to understand where, what and why. The same applies to other paragraphs like 146-150, 270-277, 300-306. The description reported in the lines from 332 to 339 should present the testing procedure adopted but it is not clear. Did you pre-crack the samples? What type of sample did you use? What type of notch, if any, have you used?
I would suggest to change the manuscript in order to present the information in a more logical way introducing the problem, discussing the literature and presenting the work. In particular, it is not clear the particular contribution the authors are presenting and the comments I am only reporting a small part of the comments I would have on this manuscript.
Author Response
Dear reviewer,
Thank you very much for your comments in order to improve our work. All of them were addressed as follows:
- Maybe due to the title and/or to no being clearly explained in the introduction, I think you misunderstood what the work is about. This work does not deal with crack growth by fatigue; the work deals with the application of FAD to sucbritical crack groth in HIC conditions, being all the test carried out at slow deformation rate in a quasi-static machine, obtaining a force-COD graph were the force increases up to a máximum load and, after some subcritical propagation, the faylure of the specimen takes place after the load had decreased. No dinamic oscilating loads were applied at all, and no fatigue life is considered either. This was clarified in the introduction.
- English quality was improved as much as it was posible. But if it is still not enought it can be send to a native English speaker in order to check it in a second revision round (as the due date was in 5 days this couldn’t be done).
- The terminology was corrected, modifiying the suggested expressions and some others.
- It is true that Kic is the fracture toughness and Kmat is the fracture parameter employed for the calculations of a certain material, that can be linear-elastic or elasto-plasctic depending on the situation. This was clarified in the text including some references to codes and standards that explain it, as well as what is Kmat in order to be applied in a FAD diagram.
- A light explanation for the methodology based on GE-EPRI was included in the text, as well as a reference of a work were it is explained in detail and the reference to figure 1, where an example of its application is presented. Explaining the whole methodoloy in detail will make the paper very long and is not its main purpose.
- The list of references was accidentally modifed during the post-processing. It was corrected. A similar thing happened with the list of tables and figures appearing “error, reference not found”; this was also fixed.
- SEM analysis from figures 14 to 17 was employed in the interpretation of the ressults, as well as P-COD curves form figure 13, but in the FAD representation from figures 18 to 21 just two of the SEM images (the most representative) were included in each case in order for the figures to be clear. It is necessary to include this figures in order to support the different mechanisms (IG, TG and tearing) if a reader wants to study the micromechanisms taking place during the whole subcritical propagation.
- It was clarified in the text what is the model “previously exposed”, and it was indicated the specific part of the background where it is explained.
- The test procedure and all the concernings marked were included in the text, under the epigraph test plan. 25 mm thick 20% side grooved C(T) specimens were used; The samples were pre-cracked by R=0.1 fatigue load according to [27], and then 20% side-grooved. After that, the P-COD slow strain tests in the environments as indicated in the text.
Round 2
Reviewer 1 Report
Authors accepted suggested changes.
Author Response
Dear reviewer,
Thank you for your time. English was checked by a native English speaker. Also, some clarifications were introduced in the text, in order to maker the paper easier to understand.
Best regards.
Reviewer 3 Report
The Authors have made changes to the manuscript but the main concerns raised in the previous revision were not addressed. I still find difficult to identify the novelty and follow the discussion.
Author Response
Dear reviewer,
Thank you for your time. English was checked by a native English speaker. Also, some clarifications were introduced in the text, in order to maker the paper easier to understand and indicate the novelty.
Best regards.